# Studies on the Comparative Response of Fibers Obtained from the Pastazzo of *Citrus bergamia* and Cladodes of *Opuntia ficus-indica* on In Vitro Model of Neuroinflammation

**DOI:** 10.3390/plants13152123

**Published:** 2024-08-01

**Authors:** Jessica Maiuolo, Federico Liuzzi, Anna Spagnoletta, Francesca Oppedisano, Roberta Macrì, Federica Scarano, Rosamaria Caminiti, Saverio Nucera, Maria Serra, Ernesto Palma, Carolina Muscoli, Vincenzo Mollace

**Affiliations:** 1IRC-FSH Center, Department of Health Sciences, University “Magna Græcia” of Catanzaro, 88100 Catanzaro, Italy; maiuolo@unicz.it (J.M.); robertamacri85@gmail.com (R.M.); federicascar87@gmail.com (F.S.); rosamariacaminiti4@gmail.com (R.C.); saverio.nucera@hotmail.it (S.N.); maria.serra@studenti.unicz.it (M.S.); palma@unicz.it (E.P.); muscoli@unicz.it (C.M.); mollace@libero.it (V.M.); 2Laboratory for Techniques and Processes in Biorefineries, ENEA—Trisaia Research Centre, S.S. Jonica 106, Km 419+500, 75026 Rotondella, Italy; federico.liuzzi@enea.it; 3Laboratory “Regenerative Circular Bioeconomy”, ENEA—Trisaia Research Centre, S.S. Jonica 106, Km 419+500, 75026 Rotondella, Italy; anna.spagnoletta@enea.it; 4Fondazione R. Dulbecco, 88046 Lamezia Terme, Italy

**Keywords:** dietary fiber, neuroinflammation, *Citrus bergamia*, *Opuntia ficus-indica*, antioxidant activity, apoptosis

## Abstract

Adhering to a healthy diet has a protective effect on human health, including a decrease in inflammatory diseases due to consuming fiber. The purpose of this manuscript was to obtain and compare two extracts based on fiber (BF and IF-C), derived from two plants particularly present in the Mediterranean region: bergamot (*Citrus bergamia*) and prickly pear (*Opuntia ficus-indica*). The parts used by these plants have been the “pastazzo” for the bergamot and the cladodes for the prickly pear. In addition to in vitro evaluations, the antioxidant activity was also measured on human neurons under inflammatory conditions. Furthermore, the extracts of interest were examined for their effects on the cell cycle and the regulation of pro-apoptotic proteins, caspase 9 and 3, induced by LPS. The results indicated that both extracts had a protective effect against LPS-induced damage, with BF consistently exhibiting superior functionality compared to IF-C. Moreover, the extracts can reduce inflammation, which is a common process of disease. By exploring this avenue, studying the consumption of dietary fiber could enhance our understanding of its positive effects, but additional experiments are needed to confirm this.

## 1. Introduction

The evidence overwhelmingly supports the idea that nutrition is a vital factor in maintaining good health. The Mediterranean diet is widely recognized and extensively researched due to its numerous health benefits across various regions [1]. The Mediterranean diet guidelines advise a high intake of vegetables, fruits, fiber, nuts, cereals, extra virgin olive oil, and legumes. At the same time, they encourage a moderate consumption of meat, fish, dairy products, salt, and red wine; finally, they recommend a reduced consumption of eggs and sweets [2]. This dietary approach has been shown to protect against degenerative and chronic diseases, by preventing oxidative reactions, inflammation, steatosis, obesity, diabetes, cancer, and cardiovascular and neurological disorders [3]. Fiber (carbohydrate polymers) is an edible portion of food that can be distinguished according to its solubility in water: the soluble fiber (including some types of hemicellulose, mucilage, and pectin) has hypoglycemic, hypocholesterolemic, and hypolipidemic properties [4], along with insoluble fiber (including cellulose and hemicellulose) which play a role in contributing to the prevention of gastrointestinal diseases [5]. In addition, fiber accelerates intestinal transit, reduces nutrient uptake, prolongs satiety, and diminishes postprandial glycemic peak and insulin response [6]. Finally, the fiber undergoes bacterial fermentation in the gastrointestinal tract, affecting the composition of the microbiota and the production of important final fermentative products, both of which can be beneficial to human health [7]. Consuming a significant amount of fruits, vegetables, cereals, and legumes is the foundation of a fiber-rich diet [8].

The inflammatory process is responsible for various physiological and pathological conditions; it is a defensive response by the body and involves multiple mechanisms to eliminate the source of disturbances that triggered it [9]. Inflammation can be caused by various factors, such as pathogenic infections, physical injuries, immune system disorders, cancer, chemical exposure, and trauma [10]. Several neurological disorders are linked to the inflammation of the nervous system, which is called “neuroinflammation”. Neuroinflammation can be caused by infections, brain injuries, autoimmune diseases, abnormal immune responses, oxidative damage stress, and many more [11]. The main protection of the brain and spinal cord from neuroinflammation is provided by two barriers known as “the blood–brain barrier”, BBB and “the blood–nerve barrier”, (BNB): they consist of a network of endothelial cells, lining the blood vessels in the brain and the peripheral nerve, preventing the passage of harmful substances and cells from the immune system into the central nervous system. BBB and BNB are also formed from other cells (pericytes, astrocytes, neurons, microglial cells) with specific protective functions [12,13,14,15]. BBB and BNB help maintain nervous system balance and reduce neuroinflammation [16]. To investigate neuroinflammation in vitro, researchers can treat specific cell lines with lipopolysaccharide (LPS), which is the primary constituent of gram-negative bacteria’s outer membrane. A robust immune and inflammatory response is observed in normal mammalian cells [17]. Recent studies have shown that plant extracts, which contain high levels of protective compounds, can effectively reduce neuroinflammation [18,19]. The intake of dietary fiber can impact neuroinflammation [20,21]. The gut–brain axis is influenced by fiber, which improves gut functioning and affects brain health [7]. Human neurons exposed to inflammatory stimuli were tested for the effects of dietary fiber from two plants. The plants in question are bergamot and prickly pear. The choice of these two specific plants was not accidental, but is justified by several fundamental reasons:(1)Both grow easily in the Mediterranean area and the close vicinity of our geographical location.(2)Bergamot provides excellent qualities, and its fruits are exported all over the world.(3)Prickly pear grows spontaneously and without having to receive special care.(4)Both are readily available and our research team has already used them several times, as demonstrated by our previous publications.

Bergamot, scientifically known as *Citrus bergamia (C. bergamia)*, is a member of the Rutaceae family and the *Citrus* genus. Nevertheless, it thrives best in a narrow coastal region in Reggio Calabria, Calabria, Italy [22], where the plant yields superior quality. This phenomenon occurs due to the ideal climate and soil composition in this area [23]. Bergamot has a diverse composition, which explains its numerous health benefits, like its anti-inflammatory and antioxidant properties [24,25]. In addition, bergamot also has anti-cholesterolemic activity, protective for the immune system and the cardiovascular system [26]. Bergamot fruits undergo industrial processing to produce bergamot juice and oil. However, the leftover solid residues from the peel, pulp, and seed fragments are combined to create a mass known as “pastazzo”, primarily utilized as livestock feed [27]. The pastazzo is composed of fibers, amino acids, fats, flavonoids, and minerals [28]. The fiber is formed by cellulose, hemicellulose, pectin, and inulin and, together with polyphenols, can exert protective effects against different cellular dysfunctions [29].

The *Opuntia ficus-indica* (*O. ficus-indica*) is a perennial succulent plant that can grow up to 3–5 m in height. It belongs to the family of Cactaceae and genus *Opuntia*. *O. ficus-indica* is the best-known plant in its genus, especially for its environmental adaptability. It is native to the Americas, where it is widespread in arid, semi-arid, tropical, and sub-tropical areas, but it is also present in Africa, Australia, and countries in the Mediterranean area [30]. The stem is formed by morphological modifications of the branches, of globose, cylindrical, or flattened shape, called “cladodes”, which can be from a few cm to 40–50 cm, depending on the species and the age of the plant [31]. The cladodes are covered with a layer of wax, which limits perspiration. In addition, water saving is also guaranteed by stomata which, contrary to the usual pattern, open at night and remain closed during the day [32]. The primary metabolites found in cladodes include water, fiber, polysaccharides, mucilages, polyphenols, fatty acids, vitamins, protein, and minerals. Nonetheless, the chemical makeup can vary and is influenced by multiple factors, including the season of growth, age of the plant, soil characteristics, and geographic location [33,34]. The presence of insoluble fiber in cladodes enhances their value in the food industry by acting as prebiotics that can positively influence the gut microbiota composition [35], making them more competitive. For these reasons, cladodes are not only nutritional promoters but are also used in the medical, pharmaceutical, and cosmetic fields [36].

A recent extremely encouraging and interesting study has shown a synergistic effect of the two extracts from *C. bergamia* and *O. ficus-indica* on lipid profile in subjects with mild hypercholesterolemia [37]. In particular, a daily intake of this nutraceutical, consisting of bergamot and prickly pear, has been shown to significantly reduce cholesterol and triglyceride levels in hyperlipidemic patients, highlighting an important role in cardiovascular health. The merit of these results can be attributed to the composition of the extracts: flavonoids, mixed with soluble fibers, plant sterols, and vitamin B. In parallel, we studied the extracts obtained from different portions of these two plants, assuming that the same composition can justify protective results in the experimental model used.

The manuscript had two purposes:(1)To extract fiber from the desired plants (from the pastazzo of *C. bergamia* and the cladodes of *O. ficus-indica*).(2)To compare their effects on a neuronal cell model subject to an inflammatory stimulus.

## 2. Results

### 2.1. Extracts Obtained and Tested

The process of extracting the fiber from the pastazzo of the fruit bergamot generated Bergamot Fiber (BF). The extraction of insoluble fiber from *O. ficus-indica* cladodes (IF-C) was carried out with an innovative method [34]. Both extractions are decrypted in Section 4.

### 2.2. Fibers Contained in BF and IF-C

Fiber analysis in the extracts of interest allowed us to evaluate the following classes of compounds (expressed as a percentage): (1) glucans, (2) xylans, (3) insoluble lignin, (4) soluble lignin, (5) ash, (6) other carbohydrates, and other compounds, as can be seen in Figure 1. The most widely present fraction was glucans, with the content significantly higher in IF-C than in BF. The percentages represented by xylans, other carbohydrates, and insoluble and soluble lignin are approximately equal between BF and IF-C. Another fraction significantly distributed differently between BF and IF-C is related to other components not detectable with this method, including polyphenols, flavonoids, lipid molecules, terpenes, and proteins: for this fraction, BF is richer than IF-C.

### 2.3. Total Polyphenols and Flavonoids

The content of polyphenols and flavonoids has already been measured in BF and F-C and represented in two previously published manuscripts [29,34]. However, they were never measured together and compared in the same paper. In Figure 2a,b, the total content of polyphenols and flavonoids in BF and IF-C is represented. A regression equation (y = 88.424x, R^2^ = 0.9652) gave the polyphenol content of 62.2 ± 7.4 and 138.1 ± 3.42 mg GAE/g for BF and IF-C, respectively. The calibration curve for quercetin and regression equation (y = 152.64x, R^2^ = 0.98) shows that the concentration of flavonoids in BF and IF-C was equal to 7.75 ± 3.24 and 17.25 ± 5.82 mg E-Q/g, respectively. As can be seen, the trend in polyphenol and flavonoid content was the same in both extracts and BF always showed a significantly lower amount than IF-C.

### 2.4. Concentrations of BF and IF-C

We selected the concentrations of BF and IF-C to be used in this experimental work and, as reported in Figure 3, viability and cell mortality experiments were carried out in panels a and b, respectively. The choice of considering both viability and cell mortality experiments was justified by the hope of obtaining results that could be overlapped with two different methods to be sure about the concentrations used. Tested concentrations of both extracts on SHSY5Y cells were 10, 100, and 1000 µg/mL. As can be appreciated, the treatment with BF and IF-C alone did not result in any alteration translating into reduced viability or increased mortality, and the values obtained were similar to those of untreated cells. Treatment with LPS caused significant damage to cells, while pretreatment with BF and IF-C, followed by exposure to LPS, reduced cell alteration: the protection was concentration-specific. On this cell line, BF exerted a more significant protection at a concentration of 10 µg/mL, while the higher doses (100 and 1000 µg/mL) gradually reduced this effect. Otherwise, IF-C reduced LPS-induced damage only to 100 µg/mL concentration and other doses (10 and 1000 µg/mL) were ineffective. In light of the results obtained in this work, BF was used at a concentration of 10 µg/mL and IF-C at a concentration of 100 µg/mL. Results of cell viability and mortality were superimposed.

### 2.5. Antioxidant In Vitro Potential

First of all, the antioxidant activity of BF and IF-C were evaluated on the obtained extracts through the tests indicated below.

#### 2.5.1. Measurement Reducing Power and Chelating Activity

Both BF and IF-C have a reducing power that is slightly lower than ascorbic acid (500 μM, 24 h), used as a positive control. However, between the two extracts considered, BF showed a statistically higher effect than IF-C. This result is highlighted in Figure 4a. In Figure 4b, the chelating activity of extracts is reported: BF and IF-C highlighted the same trend pictured in Figure 4a, and BF showed significantly more activity at all tested concentrations. In conclusion, we can state that the tests used to measure the antioxidant capacity of both extracts suggest that BF works better than IF-C.

#### 2.5.2. Oxygen Radical Absorbance Capacity (ORAC) Assay

The antioxidant properties of BF and IF-C were also measured using the ORAC assay and are reported in Figure 4c. This assay evidenced a different antioxidant rate in BF and IF-C extracts: BF had a higher antioxidant activity than IF-C and the corresponding curve is placed near the curve with a Trolox concentration of 7.6 µg/mL. In contrast, the curve of IF-C is adjacent to the curve with a Trolox concentration of 15.25 µg/mL.

### 2.6. Antioxidant Potential on SH-SY5Y Cell Line

After studying the antioxidant activity of BF and IF-C extracts in vitro, we also looked for this activity directly on the chosen cell line.

#### 2.6.1. Reactive Oxygen Species (ROS) Accumulation

The accumulation of ROS was measured through cytofluorometer analysis: Figure 5 reports that the treatment with BF and IF-C alone did not result in any accumulation of ROS compared to untreated cells, while exposure to H_2_O_2_ (150 µM, 20 min), used as a positive control, resulted in a high and significant accumulation of ROS, as demonstrated by the shift on the right of the fluorescence peak. Treatment with LPS also led to an increase in ROS production, less evident than H_2_O_2_ and about four times that of untreated cells. Pre-treatment with BF and IF-C, followed by exposure to LPS, resulted in a significant reduction in the accumulation of ROS compared to LPS alone. However, BF reduces ROS by about 40%, while IF-C only by 20%. In panel b, the respective quantification of the results. represented in panel a, has been shown.

#### 2.6.2. Malondialdehyde Levels

Prolonged treatment of LPS (already after 48 h instead of 24 h) causes damage not detectable with the accumulation of ROS, but with the production of the malondialdehyde (MDA), a biomarker frequently used for the evaluation of lipid peroxidation. As can be seen in Figure 6a, treatment with LPS for 48 h did not cause the accumulation of ROS, and this effect has been demonstrated both by the values of all treatments that overlap with those of untreated cells and by the respective quantification shown in Figure 6b. Differently, after 48 h of treatment with LPS, we detected a significant increase in MDA levels, significantly reduced only by BF extract. These results are represented in Figure 6c.

### 2.7. BF and IF-C Restore the Cell Cycle Altered after Treatment with LPS

Treatment with LPS caused an altered distribution of the cell cycle, visible in Figure 7a, as a block in phase G2-M at the expense of the G0-G1 phase. The G2-M phase constituted 22% of untreated cells and 56% of cells exposed to LPS. Pre-treatment with extracts of interest reverted the alterations of the cell cycle: BF reported the G2-M phase to 23% and these values are similar to those of untreated cells. IF-C reduced the block to 37%. Once again, BF worked better than IF-C, while BF and IF-C alone did not induce any alteration of the cell cycle. The respective quantification is shown in Figure 7b.

### 2.8. BF and IF-C in the Damage Caused by LPS

Treatment with LPS has determined genotoxic cell damage, as demonstrated by the slight but significant increase in protein expression p53, the transcription factor that regulates the cell cycle and beyond. Furthermore, activation of p53 led to a pro-apoptotic process involving the intrinsic mitochondrial pathway, as demonstrated by the slight increase in BAX expression and the concomitant reduction of the anti-apoptotic protein Bcl2. Treatment with BF and IF-C alone did not cause substantial differences compared to untreated cells; finally, pre-treatment with BF and IF-C followed by exposure to LPS resulted in a not always significant reduction of LPS-induced damage. The expression of these proteins is shown in Figure 8a, while Figure 8b has highlighted their quantification.

### 2.9. BF and IF-C Reduce the Expression of Caspase 3 and 9

Figure 9 shows the expression of caspase-9 and caspase-3 in Figure 9a and Figure 9c, respectively. As can be seen, the treatment with LPS resulted in increased expression of caspase-9a and caspase-3 (Figure 9c), as evidenced by the shift on the right of the curve compared to untreated cells. In both cases, the pretreatment with BF or IF-C, followed by exposure to LPS, reduced the expression of these proteins: more specifically, the expression of caspase-9 was reverted only slightly from IF-C and much more from BF. Conversely, the expression of caspase-3 was greatly reduced by IF-C, and pre-treatment with BF reported values almost similar to those of untreated cells. Treatment with BF or IF-C alone did not cause any variation of expression of caspase-9 and caspase-3. In Figure 9a,c, one representative experiment is shown, while in Figure 9b,d, the respective quantifications were obtained by three independent experiments.

## 3. Discussion and Conclusions

In this manuscript, BF and IF-C have been extracted from the pastazzo of *C. bergamia* L. and the cladodes of *O. ficus-indica,* respectively, and their effects have been tested in vitro and on human neurons subjected to an inflammatory stimulus. Induction of inflammation was carried out following treatment with LPS, one of the most powerful compounds capable of triggering a powerful inflammatory response [38]. The choice of this experimental model was justified by the desire to deepen knowledge of the inflammatory process, which is the common denominator of all pathological conditions [39]. Specifically, neuroinflammation is a pathological process involving the nervous system and is associated with several neurological disorders that affect the central and peripheral nervous systems [39]. Initially, neuroinflammation is a defense process that protects the brain and has beneficial effects by promoting tissue repair and removing cellular debris [40]. However, if inflammatory responses persist for a long time, they can become harmful and promote the onset of neurodegenerative diseases [41,42]. Neurodegenerative diseases (including Alzheimer’s Disease, Parkinson’s Disease, Multiple Sclerosis, Amyotrophic Lateral Sclerosis, Frontotemporal Dementia, and Huntington’s Disease, among the most well-known), determine the loss of neurons in selective areas of the brain causing cognitive, behavioral, motor, speech, swallowing, and breathing disorders [43]. The pathological mechanisms of neurodegenerative diseases are not completely known, and triggers can be genetic, environmental, or endogenous [44]. The incidence of neurodegenerative diseases has increased exponentially, and this condition represents a growing cause of mortality and morbidity worldwide. Since, to date, no drug treatment is fully resolving [45], further studies are needed to increase knowledge about this topic. For this reason, the model of neuroinflammation described in this experimental work may help to understand the underlying mechanisms that are involved [46]. As already described and confirmed by the results obtained, BF and IF-C are made of fiber (Figure 1); therefore, the consumption of these respective portions of the plant (pastazzo of *C. bergamia* and cladodes of *O. ficus- indica)* ensures a significant intake of fiber. Dietary fiber intake is markedly different in the various areas of the world: it has been calculated that, on average, adults in the United States consume 12–18 g/day of dietary fiber, in the United Kingdom 14 g/day and 16–19 g/day in Europe. On the contrary, the fiber consumed in Africa is about seven times greater [47]. Cross-sectional studies of different human populations in the world revealed that increased dietary fiber intake was correlated with a better intestinal microbiota composition, while a low fiber intake led to the depletion of the human gastrointestinal microbiota and subsequent increases in chronic diseases such as cardiovascular disease, obesity, colon cancer, and type 2 diabetes [48]. Dietary fiber intake and the resulting health of the microbiota are also strongly related to neuroinflammation [49] and, for this reason, we wanted to study BF and IF-C in our experimental model. The first important result reported was the qualitative and quantitative analysis of the fiber present in the extracts of interest. The main constituents were glucans, a heterogeneous family of beta-glucose polymers, and their representatives vary in the glycosidic bond placement. Among all glucans, cellulose is the best known and constitutes the main component of the wall of plant cells. In addition to performing a structural function, glucans are powerful activators of cellular immunity, have hypoglycemic effects, reduce the level of cholesterol, and possess antioxidant and anticancer properties [50]. In particular, the glucan content was significantly higher in IF-C than in BF. Since it has been shown that phenolic compounds can bind polysaccharides and accentuate antioxidant properties [51], we can justify the antioxidant effect of IF-C, which showed a higher content of polyphenols and flavonoids, as well as a higher amount of glucans than BF. The results obtained indicate that BF and IF-C have an antioxidant action: already in vitro experiments suggest that these compounds are characterized by a high reducing power, a strong chelating activity, and a massive antioxidant activity (Figure 4). The same result was also obtained on the cell line, where BF and IF-C reduced the accumulation of ROS induced by treatment with LPS for 24 h (Figure 5). A prolonged treatment of LPS, (48 h) no longer results in the production and accumulation of ROS, but in the production of MDA, which has been reduced by the treatment with BF and IF-C (Figure 6). The role of MDA is to signal the final stage reactions triggered by ROS and results from lipoperoxidation of membrane lipids. Oxidized lipids damage the structure of cell membranes by promoting the increase of arachidonic acid, with proinflammatory activity. The presence of MDA indicates that the lipid oxidation process has occurred, the antioxidant barriers are therefore overcome, and the damage has occurred [52]. Treatment with LPS determines, as reported, cellular damage that leads to the activation of apoptotic death [53], as evidenced by the modulation of the expression of specific proteins related to apoptosis (Figure 8 and Figure 9). Among these proteins, treatment with LPS causes a slight increase in the expression of p53, a transcription factor with the function of suppressing tumor transformation and regulating the cell cycle [54]. Presumably, the expression of p53 could justify the G2/M phase block of the cell cycle observed in our experimental model [55]. It is interesting to note that all the experiments conducted showed the same trend: pretreatment with BF and IF-C reduced the harmful effects triggered by treatment with LPS. Recently, it has been increasingly believed that dietary fiber is a functional substance with strong antioxidant activity and this feature increases its beneficial role for human health [56,57]. The antioxidant property of fiber may be justified by the presence of many bioactive compounds, but in recent decades it has been shown that dietary fiber can be associated with a polyphenolic fraction [58,59]. Our results have shown that BF and IF-C contain polyphenols and flavonoids (Figure 2); the only discrepancy with our results could be that the amount of these compounds in IC-C is significantly greater than in BF, although BF has shown a greater protective role than IF-C. This apparent contrast could be explained by the demonstrated assessment that an important fraction of polyphenols remain in the residues during fiber extraction from fruit and vegetables and for this reason, they are called “non-extractable polyphenols” [60,61]. This hypothesis could also be supported by fiber analysis that showed a significantly higher content of “other compounds” (polyphenols, flavonoids, lipid molecules, terpenes, and proteins) in BF, which for this reason may be more suitable than IF-C.

The results obtained showed that:(1)the use of dietary fiber can play an important antioxidant effect;(2)it may be interesting to use fiber against many inflammatory processes;(3)the comparison between the two extracts showed for the first time the greater effectiveness of BF than IF-C, increasing the importance of bergamot for human health.

In conclusion, the experimental model constitutes the limits of this scientific work: only an organism, equipped with a digestive system and gut microbiota, can explain the degradation of fibers and their bioavailability. A cell line alone cannot trace this pattern but, as is always the case in basic research, the results can provide new ideas, essential tips and help to choose the appropriate model. For this reason, it would be interesting and indispensable to test the effects of BF and IF-C in vivo on an organism.

## 4. Materials and Methods

### 4.1. Plant Materials

The bergamot fruit was harvested from the plant *C. bergamia* (Risso et Poiteau), in Bianco, a small town in the province of Reggio Calabria, Calabria, Italy in February 2023 (temperature 9.4 °C): latitude 15°28′11″ S, longitude 19.29°14′38″ E). After harvesting, the fruit was properly peeled and squeezed until the bergamot juice, bergamot oil, and pastazzo were obtained. The latter was washed (water ratio: pastazzo = 2:1) and ground. Subsequently, the mass obtained was centrifuged (4000 rpm, 10 min) and the solid phase was dried (hot air at 60 °C, 4 days). Finally, the dried mass was ground to obtain a powder with an average particle size of 60 meshes and this operation created BF. The taxonomic identification was confirmed by full professor Salvatore Ragusa, University “Magna Graecia” of Catanzaro. The voucher specimen was deposited in the Department of Health Sciences, University “Magna Graecia” of Catanzaro under the following accession number: *Citrus bergamia* L.: 13.

A cladode of *O. ficus-indica* (with an average degree of maturity, a weight of 325 g, and a length of 18 cm) was collected at Roccelletta di Borgia, Calabria, Italy (February 2023, Temperature: 10 °C): latitude 31°57′22″ N, longitude 12.23°21′38″ E). The chosen cladode was washed with distilled water, the thorns were removed and cut pieces of about 1 cm were obtained. Subsequently, the chosen matrix was dried via a laboratory oven (ENCO, Venice, Italy, 40 °C for 4 days). The dried pieces were pulverized with a grinder (Retsch SM 2000, Haan, Germany) and stored in hermetically sealed bags, at −20 °C until use. To obtain the fiber, the Cheikh Rouhou et al. method was adopted [40] with some modifications. The chosen solid/liquid ratio was = 1/30 (*w*/*w*). The cladode powder was mixed with water at a temperature of 120 °C for 6 h (maceration time) and under constant stirring (120 rpm). The selected parameters improved the fiber yield, as previously discovered and published [34]. At the end of the maceration time, the solution was centrifuged (6500× *g* for 10 min) five times, each time resuspending the solid phase in 100 mL of water at 40 °C). Finally, the resulting solid phase was dried in the oven at 40 °C and stored at 4 °C. The taxonomic identification was confirmed by full professor Salvatore Ragusa, University “Magna Graecia” of Catanzaro. The voucher specimen was deposited in the Department of Health Sciences, University “Magna Graecia” of Catanzaro under the following accession number: *Citrus bergamia* L.: 14.

### 4.2. Fiber Analysis

The analysis of lignocellulosic materials for sugars, lignin, and ash content was performed using the NREL protocol described by Sluiter et al. [62]. The carbohydrate content was determined through a two-step acid hydrolysis method. Initially, the sample was treated with 72% sulfuric acid at 30 °C for 1 h to break down polysaccharide chains into oligomers and monomers. This was followed by a second step involving 3% sulfuric acid at 121 °C for 1 h in an autoclave, ensuring the complete conversion of oligomers to monomers. These monomers were then analyzed using ion chromatography (HPIC) with a DIONEX DX300 chromatograph (Thermo Fisher Scientific, Waltham, MA, USA), a Nucleogel Ion 300 OA column, a refractive index ED50 detector (Thermo Fisher Scientific, Waltham, MA, USA), and 0.05 M H_2_SO_4_ as the mobile phase (40 °C, 0.4 mL/min). The acid-insoluble lignin content was measured gravimetrically by filtering the residue with Whatman GFA filters. Acid-soluble lignin was quantified using a Varian Cary 500 spectrophotometer (Varian, Palo Alto, CA, USA) at a wavelength of 205 nm. Ash content was assessed by placing the sample in a muffle furnace at 575 °C overnight. Each characterization process was carried out in duplicate and the table represents the average value with the standard deviation.

### 4.3. Cell Cultures

A cell line of human neurons (SH-SY5Y) was acquired from the American Type Culture Collection and kept in Eagle’s minimum essential medium supplemented with nonessential amino acids, 10% fetal bovine serum, penicillin (100 IU/mL) and streptomycin (100 µg/mL). The cell line was cultivated in a 5% humidified CO_2_ atmosphere at 37 °C. To differentiate SH-SY5Y cells, 10 µM of all-trans-retinoic acid for 5 days was used. When the cells reached 60% confluence, they were treated with BF or IF-C for 24 h. Alternatively, the cells were pre-treated with BF or IF- for 24h and then exposed to LPS (1 µg/mL for 24 h). Finally, in the 48 h viability experiments, the cells were pre-treated with BF or IF- for 24h and then exposed to LPS for 48 h.

### 4.4. Measurement of Cell Viability through MTT Test

3-(4,5-Dimethylthiazol-2-yl)-2,5-Diphenyltetrazolium Bromide (MTT) was used to evaluate cell viability in a colorimetric assay. The 8 × 10^3^ cells were grown in 96-well plates and the treatments were performed as indicated. The medium was replaced with a phenol-free medium containing an MTT solution (0.5 mg/mL). Finally, after 4h incubation, 100 µL of 10% SDS was added to solubilize formazan crystals, and the optical density was measured at wavelengths of 540 and 690 nm using a spectrophotometric reader (X MARK Microplate Bio-Rad, Hercules, CA, USA).

### 4.5. Determination of Total Phenolic and Flavonoid Content

The total content of polyphenols and flavonoids in the extracts was calculated using the Folin–Ciocalteu colorimetric assay and the aluminum chloride assay, respectively. Different solutions of gallic acid with different concentrations were used for the calculation of polyphenols. Subsequently, 400 UL of BF and IF-C were mixed with 0.8 mL of Folin–Ciocalteu reagent diluted 10 times. After 3 min of stirring, 0.8 mL of sodium carbonate 7% (*w*/*v*) was added and the resulting mixture was left to stand for another 2 h and stirred constantly until the color developed. The relative absorbance was measured at 760 nm with a Prism V-1200 spectrophotometer (Prism, Reston, VA, USA), and the total phenolic content of the extracts was expressed in mg equivalent gallic acid (GAE)/g dry weight. For the determination of the total flavonoid content, the colorimetric test of aluminum chloride was used, in which 1 mL of each extract was mixed with 1 mL of 2% aluminum chloride in methanol. After 30′, the absorbance at 430 nm was measured and the equivalent of quercetin per gram of extract (mg QE/g extract) was used to represent the estimated content of flavonoids.

### 4.6. Reducing Power Assay

The reducing power of the extracts was evaluated by spectrophotometric detection of Fe^3+^-Fe^2+^ transformation method, as previously reported [63], using ascorbic acid as reference standards. Several aliquots of various concentrations of the standard and test sample extracts (0.01–0.32 mg/mL) in 1.0 mL of deionized water were mixed with 2.5 mL of phosphate buffer (pH 6.6) and 2.5 mL of potassium ferricyanide (1%). The mixture obtained was incubated at 50 °C in a bath for 20 min. 2.5 mL of trichloroacetic acid (10%) was added and the solution was centrifuged (3000 rpm for 10 min). The top layer of the obtained solution was mixed with 2.5 mL distilled water and a freshly prepared 0.5 mL ferric chloride solution (0.1%). Absorbance was measured spectrophotometrically at 700 nm. An increase in absorbance of the reaction mixture indicates an increase in reducing power.

### 4.7. Ferrous Ion (Fe^2+^) Chelating Activity Assay

The chelation analysis of the ferrous ion has been obtained by the reduction of the absorbance at 562 nm of the ferrozine and iron(II) complex; this investigation is used to evaluate the antioxidant potential. Experimentally, 1 mL of sample was mixed with 1 mL of methanol and 0.1 mL of 2 mM FeCl_2_ and the reaction was started by adding 0.2 mL of 5 mM ferrozine. The mixture was kept at room temperature for 10 min and the absorbance was determined at 562 nm. Ascorbic acid has been used as a positive control.

### 4.8. Oxygen Radical Absorption Capacity Analysis (ORAC) Assay

The antioxidant activity of BF and IF-C was determined through the Oxygen Radicals Absorbance Capacity (ORAC) test. This method measures the antioxidant activity of a sample by evaluating the transfer of a hydrogen atom. Fluorescence loss of fluorescein (used as a probe) was measured over time and this fluorescence is justified by the formation of peroxylic radicals, following spontaneous degradation of 2,2′-azobis-2-methyl-propanimidamide, dihydrochloride (AAPH), which occurs at 37 °C. The peroxylic radical oxidizes the fluorescein and causes the gradual loss of the fluorescent signal. Antioxidants suppress this reaction and inhibit signal loss: 6-Hydroxy-2,5,7,8-tetramTethylchroman-2-carboxylic acid (Trolox) is a water-soluble analog of vitamin E that inhibits fluorescence decay in a dose-dependent manner and the fluorescein and AAPH solutions have been prepared in PBS (pH = 7.0) at concentrations of 0.02 mg/mL and 59.8 mg/mL, respectively. Trolox was produced in PBS (pH = 7.0) at concentrations of 7.65, 15.25, 30.5, and 61 μg/mL.

Finally, BF and IF-C were used with a concentration of 25 and 100 µg/mL, respectively. The evaluation of fluorescent decay for fluorescein was conducted using a microplate reader, with excitation and emission wavelengths of 485 nm and 520 nm, respectively. The measurements were made in triplicate every 2 min for 1.5 h, and the data obtained from the fluorescence vs. time curves are reported as the average antioxidant efficacy of the antioxidant compound. A regression equation was constructed by comparing the net area below the fluorescein decay curve and the Trolox concentration. The area under the curve has been calculated with the following equation:
$$\begin{array}{c} \;\;\;\;\;\;\;\;\;\;\;\;\;\;\;\;\;\;\;\;\;\;\;\;\;\;\;\;\;\;\;\;\;\;\;i=90\\ \mathrm{AUC}=1+\Sigma \;f1/fo\\ \;\;\;\;\;\;\;\;\;\;\;\;\;\;\;\;\;\;\;\;\;\;\;\;\;\;\;\;\;\;\;\;\;i=1 \end{array}$$


### 4.9. ROS Accumulation Measurement in SHSY5Y

The H_2_DCF-DA molecule easily penetrates cells and is cleaved by intracellular esterases to form H_2_DCF, a compound that can no longer exit cells and binds to ROS when undergoing an oxidation reaction. This last step transforms it into DCF, which is highly fluorescent. Therefore, the quantification of DCF is proportional to the content of ROS present in cells. The cells were seeded in 96-well microplates with a density of 6 × 10^4^ and, the following day, they were pre-treated as described. The growth medium was replaced by a fresh medium containing H_2_DCF-DA (25 µM) and, after 30 min of exposure to 37 °C, the cells were washed with PBS, centrifuged, resuspended in PBS, whether or not exposed to H_2_O_2_ (120 µM, 20 min) and subjected to cytofluorometer reading (FACS Accury, Becton Dickinson, Franklin Lakes, NJ, USA).

### 4.10. MDA Assay

Lipid peroxidation was evaluated by measuring the MDA content. This compound, when present, is mixed with thio-barbituric acid (TBA), forming a colorimetric product proportional to the MDA. Experimentally, the cells were plated in Petri dishes with a diameter of 10 cm, and the following day they were treated as indicated. At the end of the treatments, the cells were scraped and the resulting cell suspension was subjected to a freeze/thaw cycle. Next, 36 mM TBA was added and the mixture was heated for 60 min to 100 °C. Finally, a spectrophotometric reading was performed at 532 nm.

### 4.11. Analysis of the Cell Cycle

SH-SY5Y cells were sown in Petri dishes with a diameter of 10 cm and, after appropriate treatments, the cells were collected, washed twice with PBS, fixed with 70% cold ethanol for at least 2 h, and maintained at −20 °C. After fixation, the cells were washed with PBS, and incubated with 1 mL of PBS containing 0.5 mg/mL of RNase and 0.5% Triton X-100 for 30 min. Finally, the cells were stained with 50 mg/mL of propidium iodide and analyzed with flow cytometry using a FACS Accury, Becton Dickinson, Franklin Lakes, NJ, USA.

### 4.12. Cell Lysis and Immunoblot Analysis

The cells SH-SY5Y, grown in 10 cm plates, were washed with PBS and lysed with a preheated lysis buffer containing 50 mM TrisCl, pH 6.8, 2% SDS, and a mixture of protease inhibitors. The protein concentration in cell lysates was determined from the DCA protein assay; the samples were boiled and filled with SDS–polyacrylamide gel. After electrophoresis, polypeptides were transferred, and specific antibodies were used to reveal the respective antigens. The primary antibodies were incubated overnight at 4 °C followed by a secondary antibody conjugated with horseradish peroxidase for 1 h at room temperature. Advanced chemiluminescence procedures developed the blots. The following primary antibodies were used: a mouse monoclonal antibody for p-53 (DO-7, Thermofisher Scientific, Waltham, MA, USA) at 1:1000 dilution; a mouse monoclonal antibody for BAX (6A7, Invitrogen, Waltham, MA, USA) at 1:1000 dilution; a mouse monoclonal antibody for Bcl2 (Bcl-2-100, Invitrogen) at 1:1000 dilution and a mouse monoclonal antibody for actin at dilution 1:5000. Secondary antibody conjugated with horseradish peroxidases made in mouse was used at 1:10,000 dilution.

### 4.13. Measurement of Protein Expression through Immuno-Cytofluorometry

To measure the expression of caspases 9 and 3, monoclonal antibodies, linked to fluorochromes, were used and the samples were analyzed through cytofluorometry. The cells were washed with PBS, trypsinized, and collected in a growth medium in cytofluorometer tubes. Subsequently, they were treated with BSA in 10% animal serum and kept for 60 min at room temperature to block non-specific sites. At the end of the indicated time, the cells were incubated for 2 h at 37 °C with the desired antibodies. For caspase 9, a rabbit monoclonal antibody (2D5) was purchased from Invitrogen, at 1:200 dilution. For caspase 3, a rabbit monoclonal antibody (E87) was purchased from Abcam, Cambridge, UK (197202) at 1:200 dilution. After appropriate washing with PBS to remove excess primary antibodies, secondary antibodies, conjugated with Fitc, diluted in 5% animal serum for 1 h at room temperature, and 1:500 dilution, were added. After further washing in PBS, the cytofluorometer reading was carried out.

## Figures and Tables

**Figure 1 plants-13-02123-f001:**
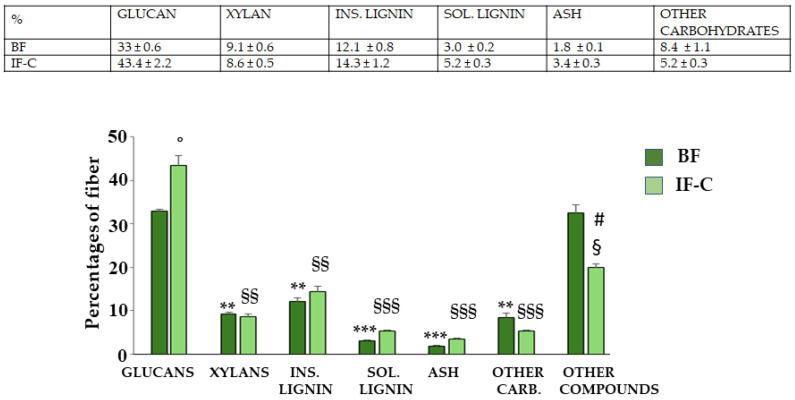
Qualitative and quantitative evaluation of fiber in BF and IF-C. Results for some fiber families (glucans, xylans, insoluble and soluble lignin, ash, other carbohydrates, and other compounds) are represented. Three independent experiments were performed, and the values were expressed as the mean ± sd. ** denotes *p* < 0.01 vs. GLUCANS of BF; *** denotes *p* < 0.001 vs. GLUCANS of BF; § denotes *p* < 0.05 vs. GLUCANS of IF-C; §§ denotes *p* < 0.01 vs. GLUCANS of IF-C; §§§ denotes *p* < 0.001 vs. GLUCANS of IF-C; **°** denotes *p* < 0.05 vs. GLUCANS of BF; # denotes *p* < 0.05 vs. OTHER COMPOUNDS of BF. A Tukey comparison test followed variance analysis (ANOVA).

**Figure 2 plants-13-02123-f002:**
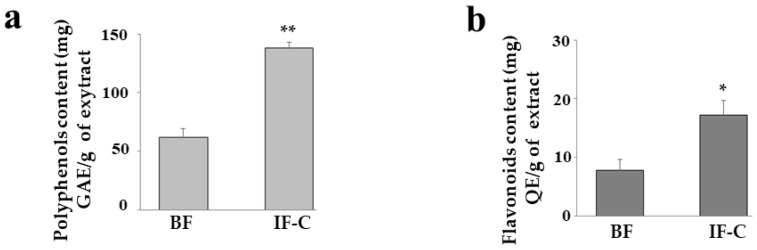
Polyphenols and flavonoid content. Folin–Ciocalteu reagent and aluminum trichloride allowed measurement of the content of polyphenols and flavonoids in both extracts. In panels (**a**,**b**), the contents of polyphenols and flavonoids are shown, respectively. Three independent experiments were performed, and the values were expressed as the mean ± sd. * denotes *p* < 0.05 vs. BF; ** denotes *p* < 0.01 vs. BF. A student’s test was applied.

**Figure 3 plants-13-02123-f003:**
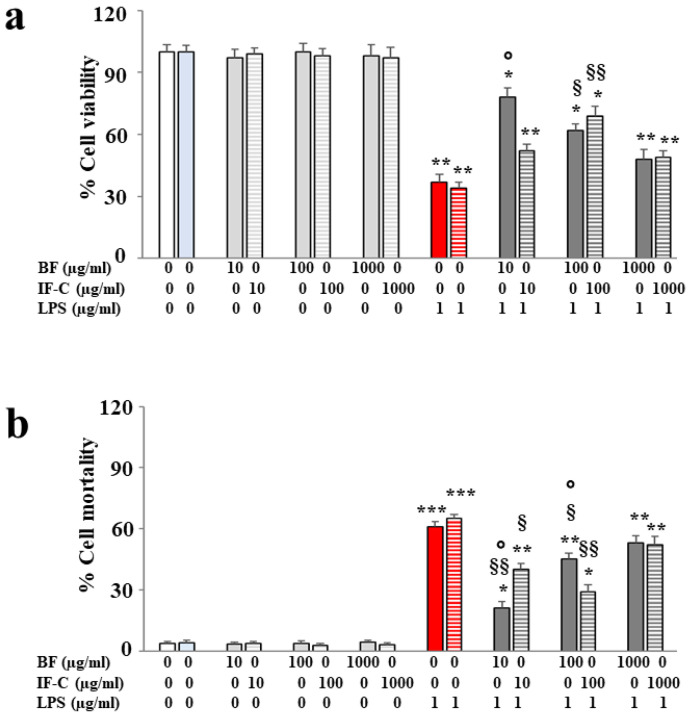
Viability and mortality of SHSY5Y cell line. Results of cell viability and mortality are reported in panels (**a**) and (**b**), respectively, following treatment with BF or IF-C b alone, LPS, and extracts + LPS. Viability experiments were conducted using the MTT test, while mortality was calculated using the Trypan blue exclusion assay. Three independent experiments were performed, and the values were expressed as the mean ± sd. * denotes *p* < 0.05 vs. CTRL; ** denotes *p* < 0.01 vs. CTRL; *** denotes *p* < 0.001 vs. CTRL; § denotes *p* < 0.05 vs. LPS; §§ denotes *p* < 0.01 vs. LPS; **°** denotes *p* < 0.05 vs. IF-C at the same concentration. A Tukey comparison test followed variance analysis (ANOVA).

**Figure 4 plants-13-02123-f004:**
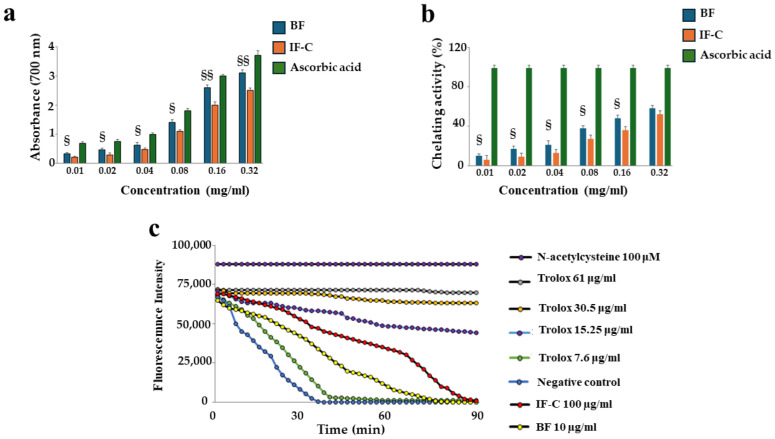
Evaluation of the antioxidant activity of BF and IF-C extracts in vitro. This figure represents the reducing power, chelating activity, and oxygen radical absorption capacity of extracts BF and IF-C, in panels (**a**), (**b**) and (**c**), respectively. Three independent experiments were performed, and the values were expressed as the mean ± sd. § denotes *p* < 0.05 vs. IF-C; §§ denotes *p* < 0.01 vs. IF-C. A Tukey comparison test followed variance analysis (ANOVA). In panel (**c**), a single representative experiment is shown.

**Figure 5 plants-13-02123-f005:**
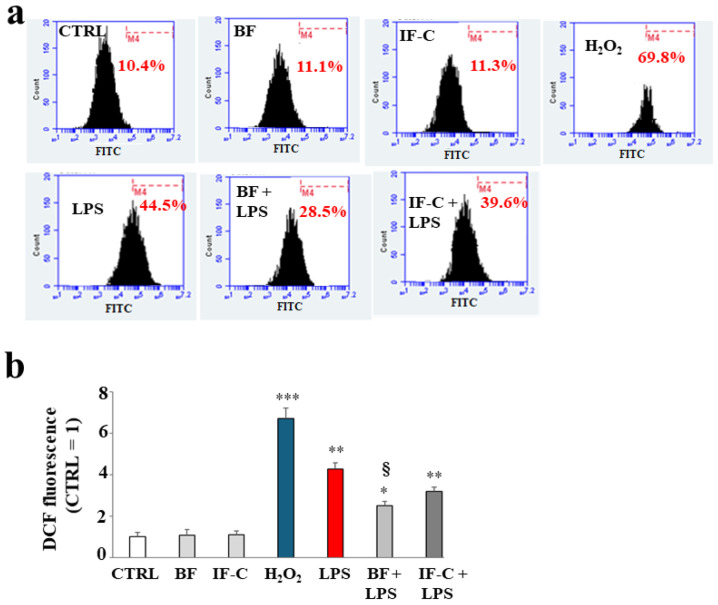
ROS accumulation. In panel (**a**), the accumulation of ROS measured with the cytofluorometer is shown. Each box refers to a treatment as indicated: the x-axis represents the fluorescence of the fluorochrome FITC connected to our fluorescent probe, while the y-axis is relative to the number of cells that we decided to acquire (10,000). At the top of each box, there is a marker (M4), which is arbitrarily drawn in the control and kept the same for all other samples. The part of the peak included in M4 is indicated by a numerical percentage. In panel (**b**), the respective quantification, obtained from comparing the percentages, is represented. The control percentage is arbitrarily made equal to 1 and the other values are related to it. Three independent experiments were carried out, and the values are expressed as the mean ± standard deviation (sd). * denotes *p* < 0.05 vs. CTRL; ** denotes *p* < 0.01 vs. CTRL; *** denotes *p* < 0.001 vs. CTRL; § denotes *p* < 0.05 vs. LPS. A Tukey comparison test followed the analysis of variance (ANOVA). The results, shown in panel (**a**), refer to a representative experiment of the three conducted.

**Figure 6 plants-13-02123-f006:**
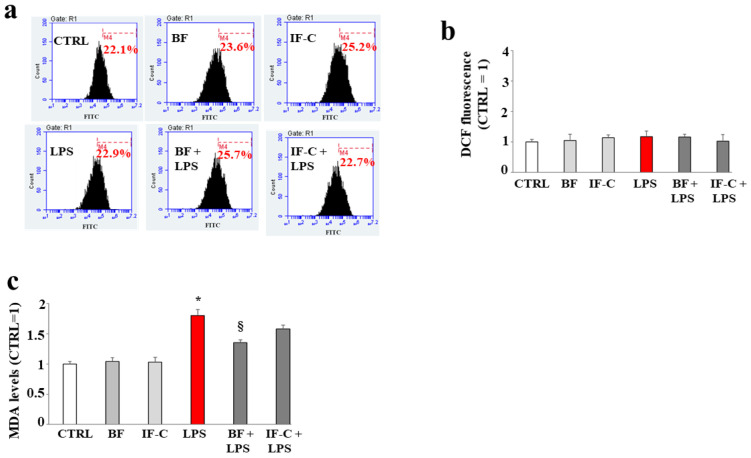
MDA accumulation. In panel (**a**), the absence of ROS, following treatment with LPS for 48 h, is shown. Each box refers to a treatment as indicated: the x-axis represents the fluorescence of the fluorochrome FITC connected to our fluorescent probe, while the y-axis is relative to the number of cells that we decided to acquire (10,000). At the top of each box, there is a marker (M4), which is arbitrarily drawn in the control and kept the same for all other samples. The part of the peak included in M4 is indicated by a numerical percentage. In panel (**b**), the respective quantification, obtained from comparing the percentages, is represented. The control percentage is arbitrarily made equal to 1 and the other values are related to it. Finally, in the panel (**c**), the MDA levels are represented. Three independent experiments were carried out, and the values are expressed as the mean ± standard deviation (sd). * denotes *p* < 0.05 vs. CTRL; § denotes *p* < 0.05 vs. LPS. A Tukey comparison test followed the analysis of variance (ANOVA). The results shown in panel (**a**), refer to a representative experiment of the three conducted.

**Figure 7 plants-13-02123-f007:**
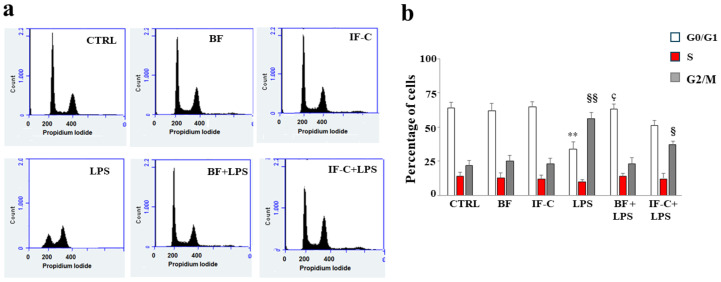
Exposure to LPS alters the cell cycle profile. The figure shows the cell cycle obtained by cytofluorometric analysis. In every single box of panel (**a**), the x-axis represents the fluorescence of the propidium iodide, while the y-axis is relative to the number of cells that we have decided to acquire (30,000). A representative experiment is shown. In panel (**b**), the respective quantification is highlighted. Three independent experiments were performed, and the values were expressed as the mean ± sd. ** denotes *p* < 0.01 vs. G0–G1 phase of the control; § denotes *p* < 0.05 vs. G2/M phase of the control; §§ denotes *p* < 0.01 vs. G2-M phase of the control; ç denotes *p* < 0.05 vs. G0–G1 phase of LPS. A Tukey comparison test followed the analysis of variance (ANOVA). The results shown in panel (**a**), refer to a representative experiment of the three conducted.

**Figure 8 plants-13-02123-f008:**
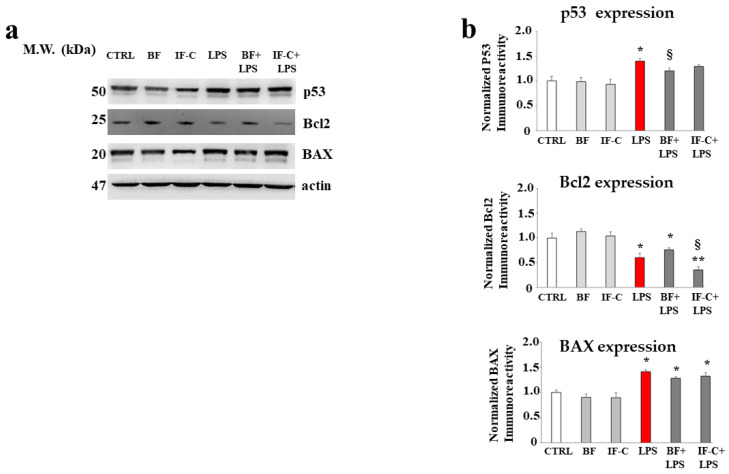
The damage induced by LPS modulates the expression of certain proteins. In panel (**a**), the expression of proteins p53, Bcl2, and BAX was represented following the indicated treatments. A representative experiment is shown. In panel (**b**), the respective quantification is highlighted. Three independent experiments were performed, and the values were expressed as the mean ± sd. * denotes *p* < 0.05 vs. the control; ** denotes *p* < 0.01 vs. the control; § denotes *p* < 0.05 vs. LPS. A Tukey comparison test followed the analysis of variance (ANOVA).

**Figure 9 plants-13-02123-f009:**
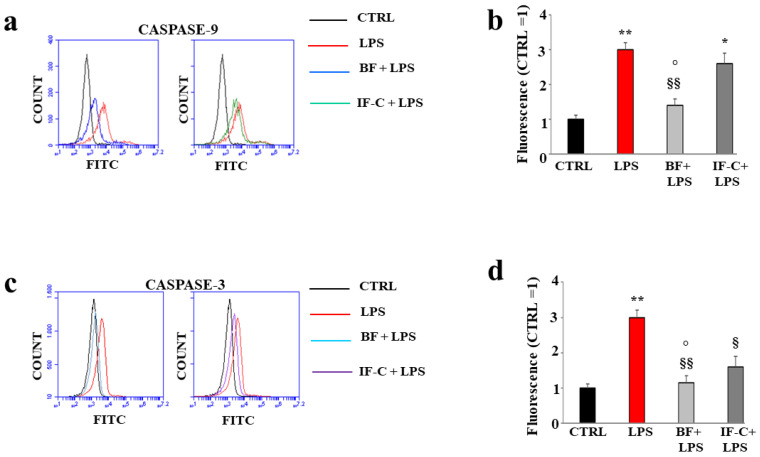
Effects of BF and IF-C on the expression of caspases. Monoclonal antibodies linked to fluorochromes were used to measure the expression of caspase 9 (panel (**a**)) and caspase 3 (panel (**c**)). A representative experiment was reported. In panels (**b**,**d**), the respective quantification is highlighted. Three independent experiments were performed, and the values were expressed as the mean ± sd. * denotes *p* < 0.05 vs. the control; ** denotes *p* < 0.01 vs. the control; § denotes *p* < 0.05 vs. LPS; §§ denotes *p* < 0.01 vs. LPS; ° denotes *p* < 0.05 vs. IF-C + LPS. A Tukey comparison test followed the analysis of variance (ANOVA).

## Data Availability

The original contributions presented in the study are included in the article, further inquiries can be directed to the corresponding author.

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
