# Peer review of "Studies on the Comparative Response of Fibers Obtained from the Pastazzo of Citrus bergamia and Cladodes of Opuntia ficus-indica on In Vitro Model of Neuroinflammation"

_plants, 2024, doi:10.3390/plants13152123_

Round 1

Reviewer 1 Report

Comments and Suggestions for Authors

This is an interesting study on the characterization of chemicals extracted from Citrus bergamia and Opuntia ficus-indica. In general, I favor the paper. However, there are several questions: 1) first of all, I could not find the rationale of the two choices of plant species. Were they just randomly selected? Please add the rationale in the Introduction.

Reviewer 2 Report

Comments and Suggestions for Authors

The fiber from plant origin provides a very important function for humans. This manuscript conducted a detailed study on the biological activity of two types of fibers that obtained from Citrus bergamia and Opuntia ficus-indica. After consuming fiber from food, it plays an important role in the physiological conditions of the intestine and the gut microbiota. However, fibers cannot be directly absorbed by the intestines and transported into the human body with functions. Only fiber degradation or metabolic products can be transported to other parts of the human body with effects. So, there are doubts about the activity of screening fibers in the effects on a neural cell model. Therefore, there are certain doubts about the significance of the topic selection in this manuscript. Exploring the biological activity of comparative fibers should be related to the physiology of the intestine, and indicators that are not related to the intestine have no significance.

1. There is no unified format, for example, when using Latin names for the second time, abbreviations need to be used. The formats of “H2SO4” and “ED50” are also incorrect.

2. The analysis of variance is not accurate. The method analysis between multiple groups was conducted using the Tukey test.

3. Figure 8 needs to be arranged more neatly.

Reviewer 3 Report

Comments and Suggestions for Authors

Dear authors

The manuscript : Studies on the  comparative response of fibers obtained from Citrus bergamia and Opuntia ficus-indica on in vitro model of neuroinflammation.

After read your extensive manuscript, I want to express that I consider that it is a very interesting work, which was well designed and the discussion and conclusions are based in your own results.

I hope to read new findings in this interesting area from your research group.

I do not have comments to change anything on your manuscript.

Reviewer 4 Report

Comments and Suggestions for Authors

The authors claim that the extracts obtained from the fibers are from Citrus bergamia, but they are from “Pastazzo”, a byproduct of the production of bergamot essential oil, which represents approximately 55% of the fruit and is composed of seeds, exhausted pulps and peels. In the title, abstract, introduction and in the rest of the text, clarify that the extracts obtained from the fibers are from “Pastazzo” and not from the fruit of C. bergamia.

Expand the background with the data published by Di Folco U, et al. A Nutraceutical phytocomplex of extracts from Citrus Bergamia and Opuntia Ficus-Indica improves lipid profile in subjects with mild hypercholesterolemia: A pilot study. Ann Med Health Sci Res. 2023; 13:816-822. Expand the background with the data published in this article, the extracts of Citrus Bergamia and Opuntia ficus-indica positively improve the lipid profile in subjects affected by mild hypercholesterolemia.

Round 2

Reviewer 2 Report

Comments and Suggestions for Authors

The authors made appropriate modifications to the manuscript. The citation of references in the manuscript is reasonable.